# Solving the Rubik's Cube in a Human-like Manner with Assisted Reinforcement Learning (ARL)

## Abstract

Human-AI collaboration is most key in situations in which AI must approach problems in a human-like manner. In this work, we present a novel approach to Rubik's cube solving that utilizes human-like solving techniques. We demonstrate assisted reinforcement learning (ARL), in which RL trains to solve the cube in separate steps (CFOP), thereby emulating human behavior.

Secondly, we applied inverse reinforcement learning (IRL) to align AI behavior with human problem-solving. We create a dataset of over 10,000 human Rubik's cube solves and train to achieve a reward function that accurately reflects the goals and preferences of human solvers. As a result, the system is able to generalize across different cube states while maintaining interpretability.

Our research demonstrates the potential of combining ARL and IRL to close the gap between human and AI behavior. We successfully highlight the interdisciplinary nature of training AI to solve a trivial task while imitating complex human behavior.

## Introduction

In recent years, the intersection of AI and human problem-solving has been a growing area of research. However, not many attempts have been made to emulate human problem-solving, which could potentially have huge benefits in enhancing interpretability, trust, and user collaboration. AI systems that resemble humans more closely are better suited for user expectations and provide solutions that are not only efficient, but more intuitive for users to follow. For example, in education, AI that solves problems in a more slow and clear manner is potentially more useful than AI that solves them in the fastest and most efficient manner.

We focus on replicating the solving of the Rubik's cube as a case study for human-like AI. By replicating the well-known CFOP to solve a cube and developing reward functions through IRL, we demonstrate the potential of human-centered AI design.

## Related Work

In the past, there have been many attempts to solve the Rubik's cube with algorithms or AI. It has been solved using deep reinforcement learning without human guidance (Roshan et al. 2024). However, these solutions tend to naturally prioritize efficiency over interpretability, picking moves that are completely unintuitive to human solvers. Other works have explored incorporating domain-specific heuristics into RL frameworks, allowing human-like solving to an extent. We build on this idea by creating distinct RL agents to solve different phases of the CFOP solving method, a technique that is familiar to human solvers.

IRL is the most common and useful tool to infer reward functions from observed behavior, making it a natural choice for tasks that require AI agents to both model human behavior and solve the problem effectively. In the context of the Rubik's cube, prior work has focused on learning optimal strategies and reward functions from human demonstrations (Abbeel 2019). We build on this by developing a custom dataset of over 10,000 human-produced solving sequences, allowing for tailored reward functions that incorporate human-like strategies and intuition into the solving process.

Interpretability in aligning AI systems with human cognitive processes has been underscored in many studies (OpenAI 2019). It has been shown that interpretability is necessary for trust and collaboration in such systems. Additionally, in AlphaZero, hierarchical decision-making to break strategies down into human-understandable steps was shown to provide numerous benefits (Cheerla 2018). To emulate such effects, we create a 3D visualization tool to display a model of the cube to convey movements through visual imagery rather than text. By adding to the interactive experience, we further increase intrepretability and trust in human-AI systems.

## Assisted Reinforcement Learning Approach

### 3D Movable Cube

A crucial component of our ARL framework was the creation of a 3D interactive Rubik's cube model. Prior studies have shown that visualization is very useful in human-centered AI, especially in tasks for which understanding every step the model takes is essential. By presenting solutions in a human-recognizable format and creating a visualization for human cognition, this was an important step to take in advancing the trust and interpretability of our system (Fig. 1).

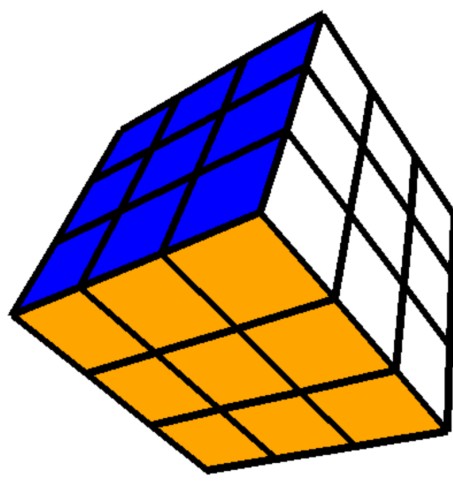

Figure 1: 3D Interactive Cube

## ARL Setup

We use a fairly standard reinforcement learning setup for the representation of the cube. Each face of the cube is represented as a 3 x 3 grid of integers, for which each integer corresponds to a color 1-6. Thus, the entire observation state is a 6 x 3 x 3 tensor. To ensure compatibility and consistency with standard RL algorithms, we flatten the state tensor into a 1D vector of length 54. This encoding provides a compact and computationally efficient representation of the cube's current configuration. To handle partial observability during training, the environment supports encoding prior states as part of the observation, forming a history of recent states.

The action space is defined as the set of all possible moves that can be applied to a particular observation state of the cube. We opt to use the twelve standard human-recognizable moves of "front", "back", "left", "right", "top", "bottom", and their respective opposite moves. Thus, each action is encoded as an integer from 0-11 to ensure seamless integration within existing RL frameworks.

$$\mathcal{A} = \{F, F', B, B', L, L', R, R', U, U', D, D'\}$$

We found that designing an optimal reward system was the hardest part of this process. This is the primary reason we opted to explore an inverse reinforcement learning approach later on. However, we observed that of the systems we tried, the proposed "Intermediate Basis" works the best.

### Intermediate Basis

- **Solved State Reward:** A large positive reward (for example $R_{\text{solved}} = +100$) is given when the cube reaches the solved state (all faces uniform in color).
- **Intermediate Reward:** A smaller positive reward $R_{\text{intermediate}} = +1$ is given for each action that reduces the Manhattan distance (Chugani 2024) between the current state and the solved state. The Manhattan distance

is computed based on the number of misplaced stickers relative to the solved configuration.

- **Penalty for Invalid Moves:** If the agent attempts an action that does not result in a valid state transition, a penalty $R_{\text{invalid}} = -5$ is applied.
- **Step Penalty:** To encourage efficiency, a small negative reward $R_{\text{step}} = -0.1$ is applied at every time step.

Additionally, we ensure actions result in valid cube transformations by verifying with a deterministic transition function. The environment terminates either when the cube is solved or after a maximum of $N_{\text{max}} = 100$ steps

**Implementation Framework** The cube environment was developed in Python, using OpenAI's Gym (Brockman et. al 2016) library to create a standardized reinforcement learning interface. We opted for this library because Gym provides a flexible framework to define action and observation spaces while facilitating integration with popular RL algorithms.

For training, we utilized the stable-baselines3 library, which has efficient implementations of many state-of-the-art RL algorithms. After rigorous testing, we found that using the Deep Q-Networks (DQN) (Raffin et. al 2021) provided optimal results.

Stable Baselines' (Mnih 2013) implementation of DQN uses two neural networks—the policy network, for approximating the action value $Q(s, a; \theta)$, and the target network, a delayed copy of the policy network that helps mitigate feedback loops caused by rapid updates of Q-values.

DQNs utilize experience replay buffers, data structures that store tuples $(s, a, r, s', d)$, which represent the current state, action taken, reward received, next state, and "done" flag respectively. Batches of experiences are sampled uniformly from this buffer to decorrelate consecutive transitions, which improves sample efficiency. This is especially important in this task because it helps break temporal correlation. Since consecutive states are highly correlated, utilizing experience replay helps produce more diverse and less biased data.

The DQN loss function aims to minimize temporal difference error (Tang et. al 2022), which measures differences between predicted Q-values and the target Q-value. Because solving the Rubik's cube is a process with sequential actions, the TD error is a good way to gauge how well agents' current actions align with future expected rewards. The full adjusted loss function we used is shown below:

$$L(\theta) = \mathbb{E}_{(s,a,r,s') \sim \mathcal{D}} \left[ \left( r + \gamma \max_{a'} Q_{\text{target}}(s', a'; \theta^-) - Q(s, a; \theta) \right)^2 \right]$$

Another appealing feature was Stable Baselines' implementation of Double Q-Learning Enhancement, to address overestimation bias in Q-value predictions. We found that without using this, models would overestimate the value of certain actions, which would lead to many cycles or repeats.

DQNs utilize an $\epsilon$-greedy for exploration, and we added a linear decay schedule for $\epsilon$, to reduce exploration over time. This helped with avoiding local optima, which was extremely common in a puzzle like the Rubik's cube. In addition, the epsilon value decays as the agent learns, which allows it to use its learned knowledge more and more.

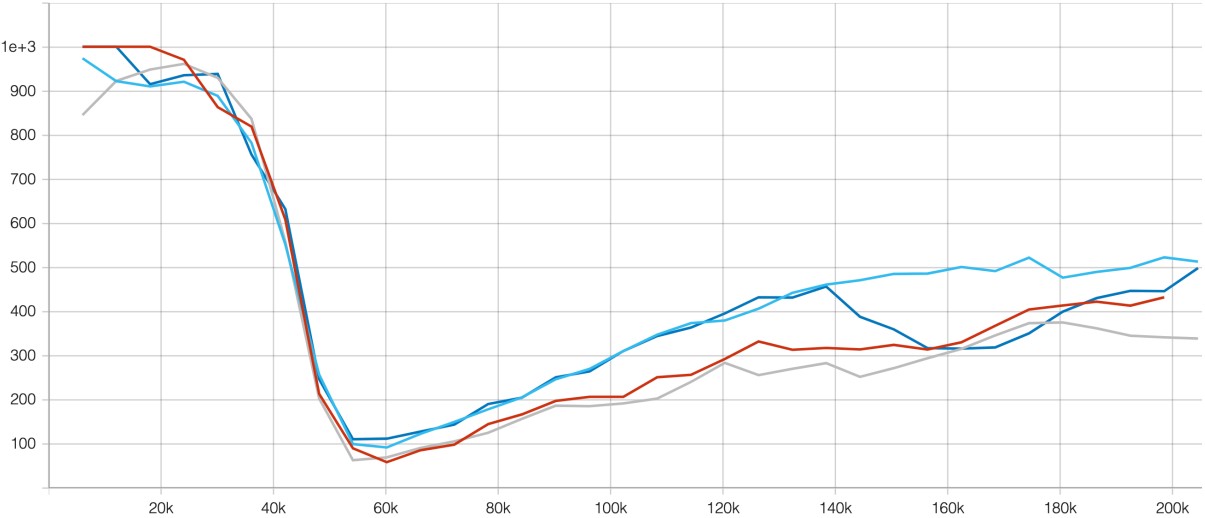

Figure 2: Cumulative Mean Episode Lengths for Different Policies with Hyperparameter Tuning for Solving the Full Cube

## Training

We use a multi-layer perceptron architecture for training (MLP Policy) (Song and Sun 2021). We employ hyperparameter tuning by using Optuna (Akiba et. al 2019), a popular optimization framework. The tuning was done on 50 trials, and for each trial, a model was trained for 50,000 timesteps. The performance metric we used was episode lengths, which was evaluated over 20 episodes for each trial. The table below shows the model hyperparameters that were randomly sampled during the tuning process.

Table 1: Randomly Sampled Model Hyperparameters during Tuning Process

| Hyperparameter | Randomly Sampled Range | Optimal Value |
|---|---|---|
| n_steps | 2048 - 8192 | 6004 |
| gamma | 0.8 - 0.9999 | 0.85 |
| learning_rate | 0.00001 - 0.0001 | 0.00009 |
| clip_range | 0.1 - 0.4 | 0.22 |
| gae_lambda | 0.8 - 0.99 | 0.91 |

After obtaining the model with the best parameters, the final PPO model was trained for 75,000 timesteps. We utilize a custom training callback for checkpointing every 10,000 steps and preventing any potential overfitting.

## Summary

These steps were repeated for each step of CFOP (the cross, the first two layers, OLL, and PLL). In the end, we were able to achieve a cumulative average of 64 moves to solve the full Rubik's cube from a random scramble (evaluated 100 times). Although this number pales in comparison to top-end algorithms like Kociemba's algorithm or even professional Rubik's cube solvers, it solves the cube in a very

basic and intuitive way. This helps users understand what is happening and makes it an appropriate guide to help beginners solve the cube. The training cycles for models of different hyperparameters are shown (Fig. 2). The light blue curve represents the model without any tuning, and the red curve represents the model with full hyperparameter tuning. The other curves are intermediate steps.

## Inverse Reinforcement Learning Approach

### Data Collection and Stratification

To model human problem-solving in the Rubik's cube, we manually collected a comprehensive dataset of over 10,000 human-produced solve sequences. We sourced the data from three primary channels.

First, we took solved sequences from over 50 professional "speedcubers", with each contributing an average of 60 sequences. We obtained this data from official competition records. On average, each sequence length was about 52 moves long.

Second, we took solved sequences from over 200 beginner solvers, with each contributing an average of 25 sequences. We obtained this data from online forums, surveys, and competition records. On average each sequence length was about 82 moves long.

Last, we took solved sequences from 10 intermediate solvers, with each contributing an average of 200 sequences. We obtained this data from willing participants. On average, each sequence length was about 68 moves long (Fig. 3).

The intention for this data stratification was not to get a sample of the most efficient solves, but rather to get a diverse selection of all Rubik's cube solvers. In this way, the final cube solver would be understandable and intuitive to solvers of all skill levels. We can thus demonstrate a successful method of increasing interpretability and user trust.

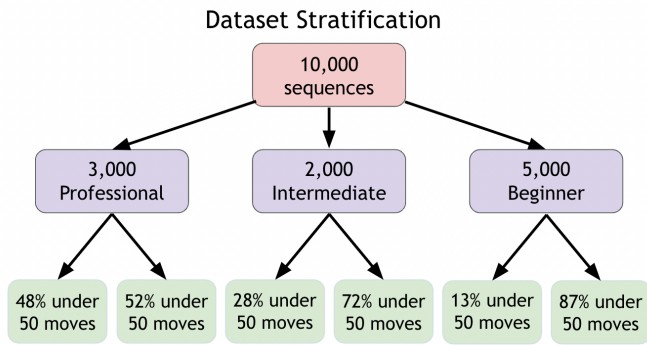

Figure 3: Data Stratification Tree

## Data Preprocessing

Each solving sequence was tokenized into a standard notation for every Rubik's cube move. We then generated state-action pairs where each state represented a cube configuration and the action corresponded to a move applied to the cube.

Because the Rubik's cube has over $4.3 \times 10^{19}$ possible configurations, the state space was very large. To combat this issue, a state hashing feature was implemented to efficiently map cube configurations to unique identifiers. This allowed us to do faster lookups and comparisons.

## Simulation Environment Setup

We built a custom environment with OpenAI's Gym to simulate cube dynamics. The observation space was encoded as a flattened vector to represent all cube stickers. The action space was defined as 12 discrete actions. We initially defined the reward structure as sparse, (+1 for solving each stage of CFOP). This would later be adjusted during the IRL training stage.

## Implementing Adversarial IRL

We opted to use adversarial inverse reinforcement learning (AIRL 2019) because of its ability to generalize well in environments with high-dimensional state spaces, such as those in this task. Unlike other classical IRL algorithms. AIRL uses GANs to jointly learn a reward function and policy. Another bonus of AIRL is that it decouples the learned reward function from the policy, making it more suited for transferability across tasks and environments.

We initialized an AIRL generator with a pre-trained PPO policy trained on synthetic environments to stabilize early-stage learning. The policy model was a neural network with two hidden layers that used a ReLU activation function.

We also created a neural network architecture for developing the discriminator (used to create the reward function). This also had two hidden layers and used a sigmoid activation function. The Adam optimizer was also used to stabilize learning rates.

During training, training alternated between generator and discriminator, switching every 10 epochs. Adjusting the number of discriminator updates per generator update also

dramatically altered the results. With hyperparameter tuning, we found the optimal value for this was 3.

Because many of the libraries we used were newer, we ran into more hurdles than when using the ARL approach. We combated mode collapse in the generator by using a diversity-promoting loss function to penalize repeated trajectories. We also had to employ learning rate decay and use entropy regularization to address potential insufficient exploration.

We also adjusted the following values with hyperparameter tuning: generator learning rate, discriminator learning rate, dropout rate, and entropy regularization coefficient. This was the extracted learned reward function after training:

$$R(s,a) = \log(D(s,a)) - \log(1 - D(s,a))$$

where $D(s,a)$ is the probability assigned by the discriminator to the expert trajectories.

The final cumulative average number of moves to solve the cube with IRL was 75 moves. Our model was not very efficient in finding the moves, but solves the cube in a very human-like way, making it a good guide for human beginners.

## Conclusions: Bridging Human-AI Collaboration in Rubik's Cube Solving

In our work, we explored the intersection of AI and human cognition through the development of two cube-solving agents that integrate RL techniques, human behavioral modeling, and principles of interpretability. Our research not only makes significant findings in AI-driven puzzle solving, but demonstrates the importance of making AI systems more understandable to human users.

The development of the custom 3D cube model was crucial to the leisure of human users. We wanted to provide a interface that maximizes human connection to ensure the greatest engagement and trust.

The incorporation of assisted reinforcement learning serves to strengthen human-AI interaction by learning from both AI training and traditional human techniques. This hybrid approach balances human intuition and ML, enabling the agent to learn to solve the cube in a human-like manner.

To further understand the reward structure of human users, we pivoted to inverse reinforcement learning, in which we allowed the agent to find a reward function for itself. In particular, using adversarial IRL helped the model differentiate between expert and non-expert moves, adding to the illusion of human decision-making. The final learned reward function provides insights into the motivations behind human actions in solving the cube, which could be explored meticulously in future work.

In conclusion, our work contributes to the goal of creating interpretable AI systems. Through techniques like IRL and ARL, we move closer to creating AI systems that can truly empower human users and enhance their problem-solving capabilities.

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
