# OpenReview forum: "Solving the Rubik’s Cube in a Human-like Manner with Assisted Reinforcement Learning (ARL)"
_AAAI.org/2025/Workshop/NeurMAD — AAAI 2025 Workshop NeurMAD Submission_

### Official Review · Reviewer_4zVV · 2024-12-20
**Interesting paper but unclear of its relevance to the workshop**

**Rating:** 4
**Confidence:** 5

**Review:**

The paper considers the problem of human-AI collaboration in solving Rubik's cube. The idea is to enhance assisted RL by first learning the reward functions from trajectories (IRL). The results demonstrate that the resulting system is able to get close to human behavior.

The paper is interesting in that it solves a challenging problem with an intuitive solution. The results look good.

The relevance of the paper to mathematical discovery and neural models is unclear. There are no algorithmic contributions as it is a methodological paper (which I do not hold against the paper). Since there are no fundamental research contributions, it becomes harder to evaluate its relevance to the workshop.

---

### Official Review · Reviewer_yKTB · 2024-12-24
**Interesting approach but lacking evaluation**

**Rating:** 4
**Confidence:** 4

**Review:**

Summary:
- This paper implemented a 3D Rubik's cube visualization and trained an agent to solve rubics cubes using DQN.
Further, it demonstrates that inverse reinforcement learning can be used to train a policy based on a dataset 10000 solving sequences of human solvers.

Strengths:
 - Implementation of an agent solving rubics cubes, using DQN
 - Collection of extensive rubics cube dataset based on many different human solvers

Weaknesses:
 - No ablations on design decisions
 - No evaluation of the main claims of interpretabilty/human-alignment/trust or anything similar
 - No comparison with baselines optimizing for shortest solving sequence (e.g. DeepCubeA)
 - Serious number of missing citations
 - The paper lacks a coherent narrative, it is not quite clear how the individual parts contribute to the overall goal

Detailed points for improvement:
- Abstract
  * Parts could be rewritten for improved clarity, for example sentences such as "We demonstrate assisted reinforcement learning, ..."

- Related work
  * Please add an explanation of what CFOP really is
  * Please add a citation to support your statement about growing research area of intersection of AI and human problem-solving
  * Mention of prior works incorporating domain-specific heuristics into RL frameworks but no citations
  * Please add a citation to support your statement about IRL being the most common and usefool tool for inferring reward functions
  * Please add a citation to support your statement of prior studies showing the usefulness of visualizations for human-centered AI
  * Please add a citation to support your statement about interpretability in aligning AI systems with human cognitive processes having been underscored in many studies, but a single citation of an online article not underscoring the previous statement
  * Please add a citation to support your statement about that it has been shown that interpretability is necessary for trust and collaboration but no citations
  * Statement about breaking strategies down into human-understandable steps providing numerous benefits in AlphaZero, but citation of an online article not underscoring the previous statement
  * The following conclusion, using a 3D visualization instead of text further increasing interpretability is not supported by previous statements.

- ARL Approach
  * Please add a citation to support your statement about prior studies having shown the usefulness of visualizations in human-centered AI
  * What part of this training setup makes up the "assisted" in assisted RL? How does it learn from traditional human techniques/human intuition?
  * You introduce the interactive Rubik's cube model, but then you don't do anything with it. How do you use it? Can you show that it helps with interpretability or trust?
  * How long was the history in the observation you have eventually used? How did you model partial observability, and is this really necessary?
  * Given your sparse explanation of the moves, I am confused as to how invalid moves could occur. Some more details on this would be beneficial.
  * How do these actions relate to CFOP? These actions seem to be the default/obvious actions of a rubic's cube, also used in previous work (e.g. https://arxiv.org/pdf/1805.07470)
  * Please add a citation for Stable Baselines3 (see https://stable-baselines3.readthedocs.io/en/master/ at the bottom for proper citation)
  * Figure 2 is missing the axis labels
  * Your approach at hyperparameter optimization is commendable, however looking at Figure 2 and no reports of performance with respect to different random seeds, the performance difference might be dominated by randomness.
  * Using a callback and regular checkpointing is useful. I assume that you then select the best checkpoint according to mean episode length? IMO, this does not prevent overfitting, but allows you to select the best model before your policy diverges
  * Weird citation for the MLP used. Also, what MLP architecture did you use in the end?
  * How many steps of random permuations did you use for the initial state? Did your agent achieve 100% success rate regardless of the number of permutations?

- IRL Approach
  * Figure 3, do you mean "over 50 moves" for the right leafs?
  * How does the gym environment used here differ from the one for ARL?
  * Please cite the paper for AIRL, not only a library (https://arxiv.org/abs/1710.11248)
  * How do you initialize the generator, on what kind of synthetic environments do you pretrain?
  * What do you mean by "The Adam optimizer was also used to stabilize learning rates"?
  * Could you provide some insights and details, ideally some quantitative analysis how you determine that the policy solves the cube in a very human like way? An option to achieve this would be to use your reward model to score the actions of your policy. You could then compare the rewards your model assign to your AIRL policy, to the ARL policy, some existing solvers such as DeepCubeA (Agostinelli, Forest, et al. "Solving the Rubik’s cube with deep reinforcement learning and search." Nature Machine Intelligence 1.8 (2019): 356-363.)
  * The dataset could be a nice contribution for future work. Please consider making it public.

- Conclusion
  * How was the 3D cube model crucial to the leisure of human users?
  * It remains unclear how human intuition was added to the ARL approach
  * It would be beneficial to mention that using IRL, you managed to build a reward function modelling human behaviour, based on your collected dataset of human moves.
  * Please actually provide some insights on the benefits of the learned reward function

Minor:
- Captions should be below the table for the AAAI style
- intrepretability -> interpretabilty
- more slow -> slower
- missing words "DQNs utilize an \eps-greedy for exploration"
- Gym has been deprecated in favor of Gymnasium (https://gymnasium.farama.org/)
- Citation at the wrong place in "Stable Baselines’ (Mnih 2013) implementation of DQN uses two neural networks"

---

### Official Review · Reviewer_DzdP · 2024-12-29
**Reviews**

**Rating:** 4
**Confidence:** 3

**Review:**

## Summary

The key argument of this paper is that, Rubik's cube solver that is built with ARL may produce solutions that are not intuitive to human players. Thus, this paper propose to incorporate human prior from 10,000 collected human solutions with IRL.

## Pros

- This paper proposes to use IRL to enforce Rubik's cube solver to produce human-like solutions.

## Cons
- This paper does not evaluate the interpretability of the built Rubik's cube solver
- For well-defined symbolic problems, such as Rubik's Cube solver and Go, human-like manner does not implies good interpretability. In particular, the reviewer would consider it more meaningful to mine new Rubik's cube solving rules from automatically built solvers.
- It would be nice if the authors could use a pre-trained LLM as an initial policy model

---

### Decision · Program_Chairs · 2024-12-30

**Decision:**

Reject

**Comment:**

 We agree with the opinions of the reviewers.